# Occupational Injury Risk Mitigation: Machine Learning Approach and Feature Optimization for Smart Workplace Surveillance

**DOI:** 10.3390/ijerph192113962

**Published:** 2022-10-27

**Authors:** Mohamed Zul Fadhli Khairuddin, Puat Lu Hui, Khairunnisa Hasikin, Nasrul Anuar Abd Razak, Khin Wee Lai, Ahmad Shakir Mohd Saudi, Siti Salwa Ibrahim

**Affiliations:** 1Department of Biomedical Engineering, Faculty of Engineering, Universiti Malaya, Kuala Lumpur 50603, Malaysia; 2Environmental Healthcare Section, Institute of Medical Science Technology, Universiti Kuala Lumpur, Kajang 40300, Selangor, Malaysia; 3Centre of Intelligent Systems for Emerging Technology (CISET), Faculty of Engineering, Universiti Malaya, Kuala Lumpur 50603, Malaysia; 4Centre of Water Engineering Technology, Water Energy Section, Malaysia France Institute, Universiti Kuala Lumpur, Bangi 43650, Selangor, Malaysia; 5Negeri Sembilan State Health Department, Seremban 70300, Negeri Sembilan, Malaysia

**Keywords:** artificial intelligence, machine learning, occupational injury, occupational safety and health, features optimization

## Abstract

Forecasting the severity of occupational injuries shall be all industries’ top priority. The use of machine learning is theoretically valuable to assist the predictive analysis, thus, this study attempts to propose a feature-optimized predictive model for anticipating occupational injury severity. A public database of 66,405 occupational injury records from OSHA is analyzed using five sets of machine learning models: Support Vector Machine, K-Nearest Neighbors, Naïve Bayes, Decision Tree, and Random Forest. For model comparison, Random Forest outperformed other models with higher accuracy and F1-score. Therefore, it highlighted the potential of ensemble learning as a more accurate prediction model in the field of occupational injury. In constructing the model, this study also proposed the feature optimization technique that revealed the three most important features; ‘nature of injury’, ‘type of event’, and ‘affected body part’ in developing model. The accuracy of the Random Forest model was improved by 0.5% or 0.895 and 0.954 for the prediction of hospitalization and amputation, respectively by redeveloping and optimizing the model with hyperparameter tuning. The feature optimization is essential in providing insight knowledge to the Safety and Health Practitioners for future injury corrective and preventive strategies. This study has shown promising potential for smart workplace surveillance.

## 1. Introduction

According to International Labour Organization (ILO), 2.78 million workers died from occupational injuries and around 374 million workers experienced non-fatal injuries, annually from 2016 until 2019. Statistically, it is predicted that about 1000 workers will be injured, meanwhile, 6500 workers will suffer from occupational diseases, and more than 7500 workers will succumb as a consequence of various exposures to dangerous and hazardous working environments [1]. Besides, workplace injury had resulted in a nearly 4% loss of the world’s Gross Domestic Product (GDP) and the loss rose to 6% in certain nations ([2]. Additionally, occupational injuries and work-related diseases impact the companies’ operation in terms of reduction of the production process, shortage of skilled manpower, and weakening the competitiveness, thus, reducing the productivity of the enterprises. To some extent, these negative repercussions of occupational accidents may significantly impact the entire community, extensively in the event of supply chain disruptions. Despite numerous countries’ persistent initiatives and policies to reduce the recurrence rate of occupational injuries, the number of occupational accidents remains high [3].

Occupational accident statistics and data are valuable; therefore, it requires reliable and robust techniques in extracting the information in the data for managing the causal factors and generating the prediction patterns of occupational injury in more efficient ways [4]. Among the linked cases of occupational accidents are those involving workers’ absences. In the event of a work-related injury, employees can take time off while being fully compensated by their employers’ worker compensation program and medical expenditures. As a result, machine learning algorithms have been deployed as a tool for optimizing and reducing operating costs to increase operational efficiencies.

There are various techniques used to develop predictive models for occupational injury outcomes, such as conventional statistical methods [5,6] and machine learning (ML) approaches [7,8]. Presently, ML models are gaining popularity and their performance prediction may outperform the conventional statistical methods due to the ability of ML algorithms to process a large amount of raw data. These have initiated the emergence of deep learning methods in predicting various outcomes, especially in the application of medical and healthcare domains such as disease prediction [9,10], medical imaging diagnosis [11,12], as well as the occupational accident outcomes [13]. In addition, ML models are reliable techniques due to their potential capacities; (i) they can handle and analyze large dimensional problems, (ii) it’s adaptable in reproducing the generation of data regardless of the complexity of the data structure, and (iii) the promising ‘prognostic and elucidative’ ability of ML, thereby, the application of ML models is compatible to forecast the workplace accidents and injuries [14]. However, the exploration of these techniques in forecasting occupational injury outcomes is still lacking and restricted [15].

There are few related studies applying ML techniques in analyzing occupational injuries; (i) Oyedele et al. in their study focused on the prediction of lost time injuries (LTI) in the power transmission and distribution projects [4], (ii) Sarkar and Maiti [3] evaluated the execution of ML models in the analysis of occupational accidents, (iii) Varghese et al. demonstrated a thorough review on the risk of occupational injuries due to heat exposure [16], and (iv) Noman et al. studied the potential of ML methods in the assessment of occupational injuries among workers in Pakistan [17]. Other related works that utilized ML techniques in predicting occupational injuries are summarized in Table 1.

Based on the previous related works, there are several types of ML methods used to predict occupational injuries such as Decision Trees (DT) Random Forest (RF), Support Vector Machine (SVM), Naïve Bayes (NB), Artificial Neural Network (ANN) and other algorithms. Although these findings contributed additional value to the existing knowledge, there is still a paucity of a comprehensive analysis of the use of ML in predicting occupational injuries and the comparison of the performance prediction of different ML models [23]. To address the inadequacies of the existing body of research, in terms of, most of the previous studies focused only on a type of industry [13,24,25], therefore, limiting the generalizability of the findings and inadequate exploration of important variables of occupational injury as an example type of injury and prevalence of affected parts of the body in model development [26]. Thus, there is a compelling need to propose a study to support the overall review of the utilization of ML models in the prediction of occupational injuries and to identify the best ML model in this research domain.

The motivation of this paper is to propose a predictive model of occupational injury severity by comparing several contemporary ML techniques using the minimal factors associated with occupational injury.

Overall, the main contributions of this study are as follows:Firstly, most of the previous related studies focused only a type of industry, however, this study differs as we analyzed a large occupational injury dataset encompassing a wide range of industrial sectors. Incorporating industry-wide data on the severity of occupational injuries into the development of the proposed model may close the gap and enhance its generalizability.Secondly, the abovementioned related studies in Table 1 have utilized many input features in producing a prediction model with higher accuracy. Though, our study presented feature optimization techniques motivated by the ability of feature importance algorithms and hyperparameter optimization. We believed that the techniques may enhance the development of the prediction model by reducing the amount of data required for workplace injury prediction and classification.Moreover, there are growing concerns from the previous research that emphasizes the development of predictive analytics to help safety and health practitioners in anticipating workplace accidents [27,28]. Therefore, this study’s findings will help the International Labour Organization (ILO) and other human-resource-related government sectors better comprehend the likelihood of workplace accidents and injuries, as well as in the planning of workplace injury prevention strategies by safety and health practitioners.

This paper is organized into 6 sections including, the introduction. The step-by-step methodology is explained in Section 2; meanwhile, the findings of performance prediction are presented in Section 3 and further elaborated in Section 4. The conclusions and recommendations for future research are in Section 5.

## 2. Materials and Methods

### 2.1. Dataset

The dataset used in this study was obtained from the United States, Occupational Safety and Health Administration (OSHA, Washington, DC, USA) severe injury reports (https://www.osha.gov/severeinjury), last accessed on 25 July 2022

### 2.2. Data Preparation

Categorical variables in this dataset are the type of industry, nature of the injury, part of the affected body, type of event, type of source, hospitalization, and amputation. The type of industry used the North American Industry Classification System (NAICS), and 20 categories were identified, such as agriculture, forestry, mining, and construction. The nature of injury has 10 categories to describe the physical characteristics of the injury, for example, surface wounds, traumatic injuries, and multiple disorders. Next, the part of the affected body consists of 8 categories. Among them is the trunk, and upper and lower extremities. The event or exposure categorized how the injury was inflicted. There are also 8 categories of events such as falls, slips, trips, and exposure to harmful substances. Last but not least, there are 9 categories of source that describe the factors that caused the injury, like tools, instruments, and machinery. These categories are pre-labeled according to the Occupational Injury and Illness Classification Manual (OIICS).

Meanwhile, columns related to ID no, dates, employers’ addresses, city, state, latitude and longitude were excluded. Other columns like inspection and secondary sources were removed due to the majority of the entries containing ‘no value’. Also, any rows with empty columns were eliminated. The textual narrative column is excluded as this study aim to work on the structured data only.

In this study, only top labels by OIICS are utilized. For example, one of the top labels for the type of event is contact with objects and equipment (E06) and within this category, it expands into several sub-labels like needlestick (E61) and stuck by objects or equipment (E62). Nonetheless, to prevent the scarce representation for each criterion [29], only top labels are used for further analysis. Also, the non-classifiable class in part of the affected body, type of event, and type of source are re-categorized into ‘Other(s)’. Overall, a total of 66,405 structured data were used as the inputs in predicting the severity of the occupational injury.

The data distributions of the utilized variables are shown in Table 2 and Figure 1 illustrated the percentage of affected body parts of the occupational injuries from January 2015 until July 2021.

In executing this study, five categorical variables were chosen as the input for the model development. There were (i) the type of industry, (ii) the affected body parts, (iii) the nature of injury, (iv) the source of injury, and (v) the event of the injury. The target outcome of the study is to predict the likelihood of occupational injury severity, whether the worker is hospitalized or had an amputation.

### 2.3. Data Pre-Processing

Data pre-processing is an essential step in the development of machine learning models. If the data collected comprises out-of-range values or missing values, it can mislead the performance prediction of the models. In this study, a total of 295 (0.4%) rows with empty columns were removed and the StandardScaler function was utilized for data standardization. 

### 2.4. Data Splitting

Then, the dataset is split into two sets: (i) the training set and (ii) the test set. In this study, a 70:30 ratio, in which 70% of the data was the training set and 30% was used as the test set. The 70:30 ratio is commonly used in various studies related to machine learning classification, and this splitting ratio is believed to produce good accuracy and prevent overfitting [30]. The flowchart of the proposed methodology is illustrated in Figure 2.

### 2.5. Predictive Modeling

For the experimentation, five different machine learning algorithms were compared: Support Vector Machine (SVM), Naïve Bayes (NB), K-Nearest Neighbors (KNN), Decision Tree (DT), and an ensemble method, Random Forest (RF).

#### 2.5.1. Support Vector Machine

SVM can generate the best generalizable decision boundaries for data classification. In this algorithm, the original feature space is transformed into a space with a higher dimension based on a kernel function defined by the operator. It then separates the two classes with a hyperplane and optimizes support vectors to extend the margin between the two classes. A hyperplane is defined as a boundary that separates the two categories. The size of the hyperplane is determined by the number of input variables in the dataset [31].

#### 2.5.2. Naïve Bayes

In the NB classifier, the input features of vector *x* are expected to be statistically independent. It computes the conditional probability for each feature and then multiplies them together. One of the advantages is the NB classifier can process large-scale and high-dimensional data for prediction and classification tasks effectively. This classifier is represented as:(1)$$p\left(\omega |x_{1},\ldots x_{n}\right)=p\left(x_{1}|\omega \right)\cdot p\left(x_{2}|\omega \right)\ldots p\left(x_{2}|\omega \right)p\left(\omega \right)$$

#### 2.5.3. K-Nearest Neighbors

KNN is a method extensively used for data mining. The method will determine the similarity between the new data and available data and group the new data into the most similar categories to the existing data. The algorithm work by [32], (i) selecting the number of K of the neighbors, (ii) computing the ‘Euclidean distance’, which is to measure the distance between any two points. The formula as in Equation (2), and (iii) from the calculation in (ii), the category for a new data point is assigned to the maximum number of neighbors.
D = √((x2 − x1)^2^ + (y2 − y1)^2^)(2)

#### 2.5.4. Decision Tree

The basic components in DT are as follows: (i) the root node is the initial point of the DT model, (ii) the decision node is in charge of decision-making and extends the model into multiple branches, and (iii) the leaf node is the outcome from those decisions [33]. The classification in DT starts with the splitting of the root node into the leaf node. The splitting continues until it reaches the leaf node. At each node, the classifier selects the feature and corresponding feature threshold to execute a split. There is a maximum decrease in entropy or impurity of the dataset after the split. When the leaf only contains samples from one class, it is said to be the best-case scenario during the splitting process. To simplify the process, in DT, the training dataset is processed by the classifier to generate a tree-like decision structure, in which the starting point is a root node, and the finishing point is some leaves. DT is commonly used in the prediction analysis of occupational accidents due to its easier interpretability [34].

#### 2.5.5. Random Forest

RF is an ensemble algorithm that uses bagging as the ensemble method and decision trees as an individual method, thus helping to reduce variance and bias in improving the findings [11,13]. The classifier collaborates several decision trees and a more robust classifier with better generalization and easier to tune the hyperparameter to overcome overfitting issues [35]. For classification tasks in RF, each tree provides a classification or considers a ‘vote’. The forest then decides the classification with the majority of the ‘votes’ as illustrated in Figure 3.

In this study, the models are developed and customized according to the following configurations:Naïve Bayes: GaussianNB()Support Vector Machines: SVC (kernel = ‘rbf’, random_state = 0)Decision Tree: DecisionTreeClassifier (criterion = ‘entropy’, random_state = 0)K-Nearest Neighbors: KneighborsClassifier (n_neighbors = 5, metric = ‘minkowski’, *p* = 2)Random Forest: RandomForestClassifier (n_estimators = 50, criterion = ‘entropy’, random_state = 0)

Then, the dataset was imported into the Python environment and the following Python libraries were applied in this study:Numpy (np) is a package for scientific computing and it has time-efficient array processing capabilities [36].Pandas (pd) is an important and powerful tool for data writing, data reading, data analysis, and manipulation [37].Matplotlib (plt) is a visualization package in python. It helps to create interactive figures and informative visualization of data.Sklearn is a robust library for machine learning, especially on predictive analysis. It provides various tools including classification, preprocessing, clustering, regression, and dimensionally reduction. These machine learning algorithms were developed in the Sklearn of Python libraries.

### 2.6. Machine Learning Models Evaluation

A confusion matrix is a technique used for model evaluation, especially for classification algorithms. It is visualized in a tabular way; each row represents an actual class, meanwhile, a predicted class represents each column. From there, the counts on “True Positive” (TP), “True Negative” (TN), “False Positive” (FP), and “False Negative” (FN) are used to compute the performance metrics in assessing the models. Figure 4 is the confusion matrix. For example, in a prediction of hospitalization in this study, the definition of the confusion matrix is as follows:(TP) is the positive instances of injured workers that are actually hospitalized and correctly predicted as hospitalized.(FP) is the negative instances of injured workers that are un-hospitalized but wrongly predicted as hospitalized.(FN) is the positive instances of injured workers that are hospitalized but wrongly predicted as un-hospitalized.(TN) is the negative instances of injured workers that are actually un-hospitalized and also, correctly predicted as un-hospitalized.

**Figure 4 ijerph-19-13962-f004:**
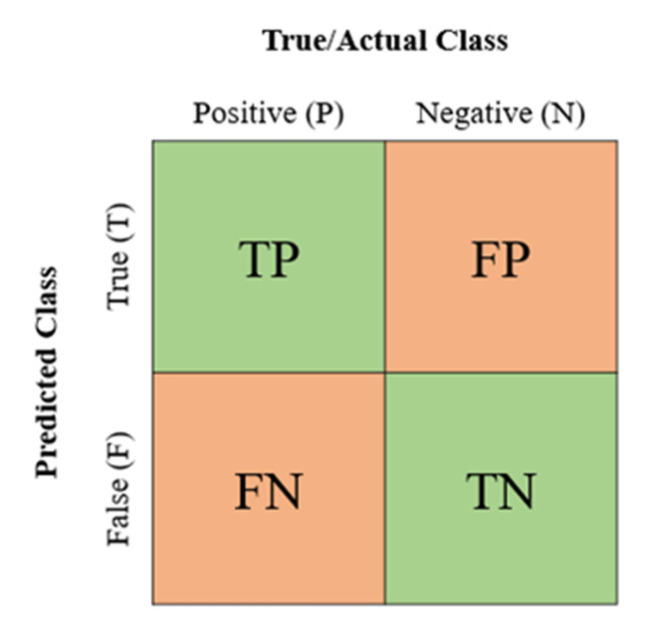
Confusion Matrix.

#### Performance Metrics

Five performance metrics; accuracy, precision, recall, F1-score, and AUC value were employed to understand and interpret the performance prediction of the machine learning models. In each metric used in this experiment, the scores ranged from 0 to 1, in which a +1 score represents model perfection [29].

Accuracy is measuring the fraction of the total samples correctly classified. It is expressed as “(TP + TN)/(TP + TN + FP + FN)”. For example, it is a ratio of correctly classified injured workers and hospitalized (TP + TN) to the total number of injured workers.Precision is the proportion of correctly classified injured workers and hospitalized to the total workers predicted to be hospitalized. It is calculated by “(TP)/(TP + FP)”.Recall or sensitivity is known as measuring the fraction of all positive samples that are correctly predicted as positive and expressed as “(TP)/(TP + FN)”. In this study, it is a ratio of the correctly classified injured workers and hospitalized divided by the total number of injured workers and hospitalized.F1-score is the “harmonic mean” of precision and recall. It is obtained by combining precision and recall into a single measure and expressed as:


(3)
$$F1-\mathrm{score}=2\times \frac{\mathrm{Recall}\times \mathrm{Precision}}{\mathrm{Recall}+\mathrm{Precision}}$$


Receiver Operator Characteristic (ROC) is extensively used to provide illustrative information on the performance of the ML algorithms. It contains information on a series of thresholds and is summarized in a single value by the ‘Area Under the Curve’ (AUC)

### 2.7. Feature Optimization

The purpose of this phase is to assess and rank the most important attribute of the occupational injury severity prediction model. Each ML model’s performance was compared, and the model with the best results was utilized to derive the important feature for occupational injury severity. The feature with the highest significance score is the most significant predictor of the model. The steps to calculate the feature importance are further elaborated in Section 3.3, depending on the best performance model algorithms.

The stage of feature optimization consisted of redeveloping the optimum performance model using only the three most important features as the input variables. The model then undergoes hyperparameter tuning, using the k-fold cross-validation technique. K-fold cross-validation is a technique used to validate the effectiveness of a proposed prediction model. The steps of how k-fold works are as follows: First, a dataset is split into a ***k*** number of folds. In the first iteration, Fold 1 is used as a testing set and the other folds, such as Fold 2, 3, 4, …, *K* as the training set. In the second iteration, Fold 2 is the test set; meanwhile, the remaining folds are the training set. This process remains until each fold has been used once, as a test set. In this cross-validation, each entry is served for validation for one time in the whole process [38].

This study employed a k-value of 10 with a number of iterations of 100, in optimizing the proposed model. The use of k = 10 is common in the applied ML model, as its practicality reduces a test error rate from higher bias or variance [39]. Theoretically, the difference in size between the training set and the re-sampling subsets will decrease as the k increases. Concurrently, the bias of the techniques is lesser as this difference becomes smaller [40]. Afterward, the cross-validation accuracy scores are computed for all hyperparameter combinations. The average cross-validation accuracy score will then be compared with the initial model with 5-feature inputs. Finally, the model with the best accuracy score is chosen as the final model. To note, in sklearn, this stage is conveniently handled by the RandomizedSearch CV method.

The proposed step-by-step feature optimization process is illustrated in Figure 5.

## 3. Results

This section is aimed to forecast the severity of occupational injuries in terms of the possibility of hospitalization and amputation. Five ML models, including SVM, KNN, NB, DT, and RF, were executed to develop the predictive systems. The binary classification was involved in the predictive system as the outcome variables are consists of two classes only, either Yes or No.

### 3.1. Performance Prediction

The performance prediction of each ML algorithm was analyzed and compared to select the best-employed model for the prediction of occupational injury severity. The findings are in two parts; the first part is the comparison of the prediction performance of SVM, KNN, NB, DT, and RF in predicting the likelihood of hospitalization, and the second part is the comparison of these models in predicting the likelihood of amputation.

#### 3.1.1. Hospitalization

In predicting the likelihood of hospitalization, all ML models used in this study had promising performances in each metric; accuracy, precision, recall, F1-score, and AUC. Specifically, RF had the best overall accuracy score of 0.89. In terms of the F1-score, RF also achieved the best performance of 0.928, similar to the DT model. Next, the ML algorithm with the highest precision was NB with 0.985, followed by SVM (0.984) and DT model with 0.98. For recall, the KNN model received the best score of 0.895. Meanwhile, the AUC value for all ML models showed significant performance ranges from 0.86 to 0.91.

For the prediction of the likelihood of hospitalization, RF was suggested as the best performance as the model achieved the highest accuracy and F1-score and the least performance model was NB as the model recorded the lowest score for accuracy, recall, and F1-score as compared to other algorithms. Table 3 shows the overall accuracy, precision, recall, F1-score, and AUC of the ML algorithms for the prediction of hospitalization.

#### 3.1.2. Amputation

Next, for the prediction of an amputation, RF showed the highest accuracy score (0.949), DT was the best precision (0.861), and the SVM model achieved the best recall (0.967) among these five algorithms. In addition, for the F1 score, RF achieved the highest score of 0.909, followed by DT and KNN with 0.907 and 0.902, respectively. All models had an AUC value of 0.95, except the NB model of 0.94.

In terms of predicting the severity of an amputation, RF was indicated as the best and most reliable model compared to the other algorithms. RF had outperformed other models as it achieved the highest score in accuracy and F1-score, as well as generated consistent scores in precision, recall, and AUC value. Nevertheless, the NB model was considered the poor performance model for this prediction, as it scored the lowest for accuracy, precision, recall, F1-score, and AUC value. Table 4 displays the overall performance of accuracy, precision, recall, F1-score, and AUC for the prediction of an amputation.

### 3.2. Performance Comparison

As the overall accuracy result is the most frequently used performance measure for classification tasks [41,42,43], the accuracy score of each ML model in both injury severity prediction, hospitalization, and amputation, are compared in determining the best prediction model for this study. It is verified that RF comparatively performs well in accuracy as compared to SVM, NB, KNN, and DT models. A pictorial representation of the accuracy performance of each model is depicted in Figure 6.

### 3.3. Feature Optimization

RF, as the best performing model, is utilized to analyze the important variables in predicting the severity of occupational injury. Attributes with the highest importance value are justified as the most significant contributor to the developed model. The calculation of feature importance through a random forest algorithm is done by the following steps [44];
(1)The individual nodes’ importance per tree is calculated using the formula in Equation (4), where $ni_{j}=\;$ importance of node j, $w_{j}$ = weighted samples reaching node j and $C_{j}$= impurity value of the node.
(4)$$ni_{j^{\;=\;}}w_{j}C_{j^{\;-\;}}w_{left\left(j\right)}C_{left{\left(j\right)}^{-}}w_{right{\left(j\right)}^{-}}C_{right\left(j\right)}$$(2)After the nodes’ importance is calculated, the feature importance per tree is determined through Equation (5).
(5)$$fi_{i}=\frac{\sum_{j:node\;j\;splits\;on\;feature\;i}ni_{j}}{\sum_{k\in \;\;all\;nodes}ni_{k}}$$(3)The calculation is normalized as per Equation (6) to a value from 0 to +1.
(6)$$normfi_{i}=\frac{fi_{i}}{\sum_{j\in \;\;\;all\;features}fi_{j}}$$(4)The calculation from (3) is averaged across the entire forest and divided by total trees by using Equation (7).
(7)$$RFfi_{i}=\frac{\sum_{j\in \;\;all\;trees}norm\;fi_{ij}}{T}$$(5)The final value is arranged in descending order, in which the most important feature appears in the first rank. The higher the value, the more important the feature.

For this experiment, the feature importance values revealed the ‘nature of injury’ as the most important variable, followed by ‘type of event’ and ‘affected body part’. The calculated values for feature importance are shown in Table 5.

Based on this finding, the model experimentation using the RF algorithm was re-developed by eliminating the 2 least important features; ‘source of injury’ and ‘type of industry’. After hyperparameter tuning, we have identified that the accuracy score of the optimized RF model was improved to 0.5%, 0.895 and 0.954 for both predictions, respectively. The cross-validation scores and hyperparameter tuning performances are presented in Table 6 and Table 7. 

## 4. Discussion

In general, our study shows that the RF model has performed better than other classifiers in terms of its accuracy. The finding is in agreement with the study by [18] where RF attained an accuracy of 91.98% in classifying employers with higher fatality risk at construction sites. It is proven in their study that RF outperformed LR, DT, and AdaBoost. In another study, [45] preferred the RF technique in predicting industrial accidents, which gave them an accuracy score of 79%. Next, the RF algorithm was executed in predicting the type of occupational accidents during construction [46]. Interestingly, their study was able to integrate the environmental data and occupational accident inputs in developing a model with 71.3% accuracy. On the other hand, the RF was observed as the most functioning model in the multi-class classification task of predicting occupational injuries including the prediction of causal factors of occupational injuries [47].

By the findings, this paper strongly recommended the ensemble method as the machine learning technique of choice, as it has demonstrated more accuracy in predicting the severity of occupational injuries. The utilization of the ensemble method in the predictive analysis is beneficial, as it collaborates the prediction of several classifiers by combining a series of weak classifiers into a single stronger classifier, thus enhancing the performance prediction. As the RF model follows the ‘majority votes decision rule’, the combination of these results will give a good generalization, therefore, resulting in higher accuracy. In terms of the F1-score, the RF model gained the highest as the model obtained higher precision and recall, as well. In principle, the higher the precision and recall, the higher the F1-score; and the higher the F1-score, the more robust the classifier [48]. Since the usage of RF-based ensemble learning is relatively limited and lacking in occupational injury studies [49], the findings are believed to support the ensemble of trees in providing more efficiency and accuracy in performance prediction, especially for data classification problems. 

In addition, the feature importance was assessed to identify the significant factors related to occupational injury. Feature importance is considered the most current strategy for assisting ML model developers to comprehend and interpret their models. Most notably, this technique is critical in providing the classification tasks with insight knowledge [50].

In this study, an RF algorithm was employed to quantify the relevance of features. As supported by [50], the RF model, as compared to Logistic Regression, proven better in explaining the feature importance in the classification models. This research has identified the ‘nature of injury’ as the most influential variable in the dataset. This variable was deemed significant by [20,31] in the mining and agriculture industries, respectively. The most prevalent types of nature of injury in the dataset were ‘open wound’ and ‘traumatic injuries to bones, nerves, spinal cord’. Next, the ‘type of event or exposure’ was the second most important feature. It is interesting to note that ‘contact with objects or equipment’ and ‘falls, slips, trips’ were the highest reported event or exposure that resulted in occupational injury.

By revealing these significant variables, it will be beneficial to the top management to design and systematically improve their ‘Workplace Injury Control Plan’ such as addressing appropriate work-safety training programs, providing adequate engineering control and personal protective equipment, as well as, maintaining the housekeeping and hygiene of the workplace environment in reducing the accident cases and lessening the severity of workplace injuries. For instance, if workers are required to handle machines, and their upper extremities are exposed, it is recommended that they be given adequate personal protective equipment and instructed on the safe operating methods for handling the machines. In addition, if employees have fallen or slipped on the job and sustained injuries to their lower extremities, frequent workplace audits and inspections, as well as proper housekeeping, are the recommended control measures.

It is important for the models to not only anticipate the severity of the occupational injury of a worker but also to include these variables, particularly the nature of the injury and how the workers were exposed into the justification that are; comprehensible and quantifiable. As indicated in Figure 7, this will enhance the models to provide the perfect future strategies for corrective and preventive measures for Safety and Health Practitioners. The applicability of a predictive model or system will be determined by the elaboration of what has to be changed for improvement and the foresight of the potential hazards and risks of workplace injuries [51].

In addition, the focus of this paper’s feature importance selection is to support the burgeoning field of study known as “Explainable Artificial Intelligence” (XAI). XAI is the technique used to explain ML predictions and aid in decision-making, particularly when these approaches are implemented in high-criticality sectors, such as medical and personal health applications [52]. In the field of occupational injuries, it is evident that worker safety, health and well-being are of the utmost importance. Detecting the severity of occupational injuries is crucial to the recovery or rehabilitation phases, as well as the injured workers’ successful return to work. Therefore, it is necessary to construct model predictions and conclusions that are explicable and interpretable to justify their trustworthiness.

Lastly, our investigation validated the need for feature optimization procedures in developing an accurate prediction model. This technique has been shown to be able to choose the most significant variables connected to the desired outcomes and eliminate fewer important variables, hence enhancing the capability of developing a high accuracy and precise prediction model using fewer variables. It is considered that prediction models with fewer variables are favored, as compared to the models with a large set of variables [53]. This is because the simpler model will ease the practitioners and operators in the field to interpret and implement in their practices [54]. On the contrary, the use of many variables is impractical due to the following reasons; (i) a large set of variables, commonly has a ‘negligible effect’ on the target outcomes [53], (ii) in terms of practicality, many variables tend to increase the computational tasks and complexity of the model development [40], and (iii) more variables in the model make the model highly dependent on the ‘observed data’, however, the data is most likely unavailable and difficult to collect. According to that, this study can highlight feature optimization as a useful technique in selecting fewer important variables in developing a prediction model with higher accuracy.

Despite producing promising performance prediction results, this study has some limitations. First, the dataset contains no socio-demographic information about the affected employees. Therefore, the investigation of the role of age, gender, and years of experience, including the job position, is restricted in this work. Next, it shall include additional Occupational Safety and Health (OSH) analytic data such as safety and health audit reports, medical information such as days off until return to work, and risk assessment results to improve the predictive capabilities of the models.

## 5. Conclusions

In this research, we demonstrate the execution of five sets of ML algorithms: SVM, KNN, NB, DT, and RF in classifying the occupational injury severity. We find that these techniques provide satisfying performances to the predicted classes, hospitalization or amputation. For both predictions, the RF model consistently outperformed other ML models with higher accuracy, F1-score, and AUC value. Consequently, this finding is essential in highlighting the potential of the ensemble learning method as a better prediction model.

For feature optimization, it has revealed the ‘nature of injury’, ‘type of event’ and ‘affected body part’ as the three most significant factors behind the prediction of occupational injury severity. After hyperparameter tuning, the accuracy of the optimized RF model is improved to 0.895 and 0.954 for both predictions, respectively. This information is beneficial for the Safety and Health Managers to continuously improve their Occupational Safety and Health Management System (OSHMS) especially in reducing workplace injury cases. Finally, this study has employed a broad section of industrial sectors as the input classification. The integration of various industries’ information may improve the generalizability of the prediction model. To the best of the author’s knowledge, this study has provided the latest baseline findings in the prospect of utilizing the feature importance and hyperparameter optimization in the prediction of occupational injury severity. In addition, the findings revealed the predictive ability of the proposed model is improved.

The goal of this research field is to incorporate the unlabeled data from the occupational injury report such as text narratives and injury images with the labeled data to improve the predictive capabilities of the model [55]. As suggested by [56], the use of the deep learning method using the Generative Adversarial Network (GAN) can assist in medical diagnosis with full utilization of labeled and unlabeled data. Moreover, [57] used the GAN-driven approach and was able to predict the patient’s length of hospitalization in managing the health resources. In relation to the occupational safety domain, [20] has proposed the GAN technique to overcome the data imbalance issues in the occupational injury report, and to investigate alternative forms of deep neural architecture.

## Figures and Tables

**Figure 1 ijerph-19-13962-f001:**
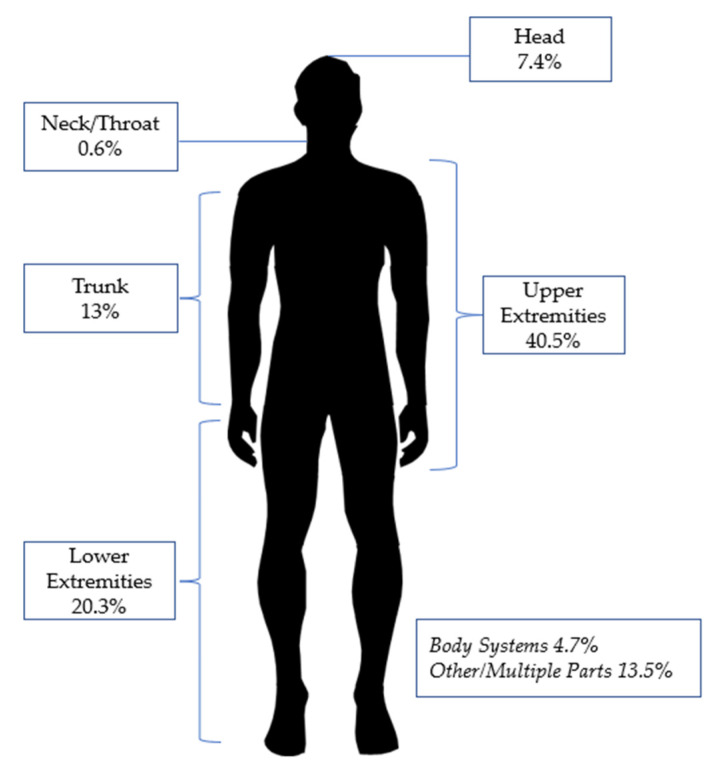
Percentage of the affected body part(s).

**Figure 2 ijerph-19-13962-f002:**
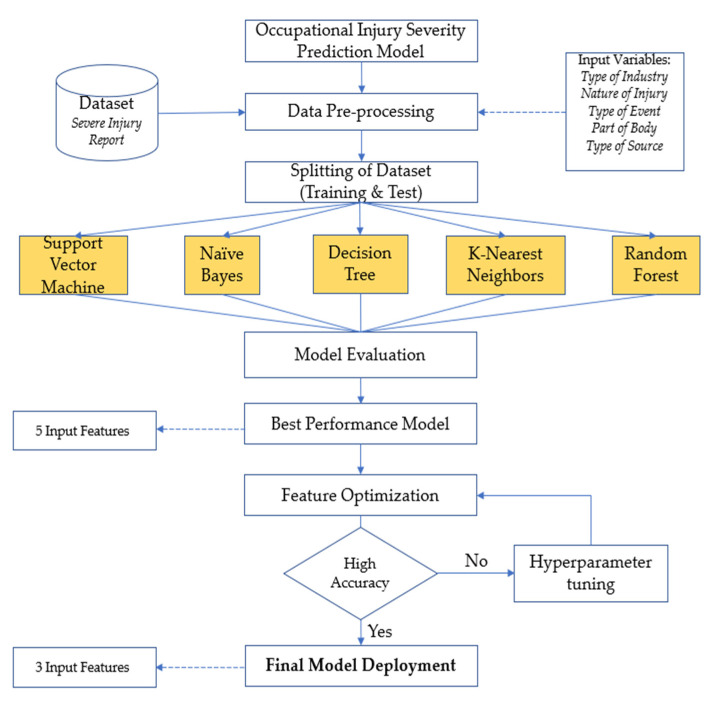
The overall research methodology.

**Figure 3 ijerph-19-13962-f003:**
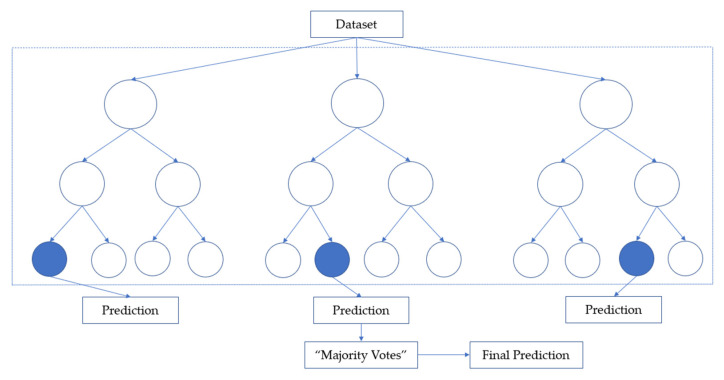
Random forest classifier.

**Figure 5 ijerph-19-13962-f005:**
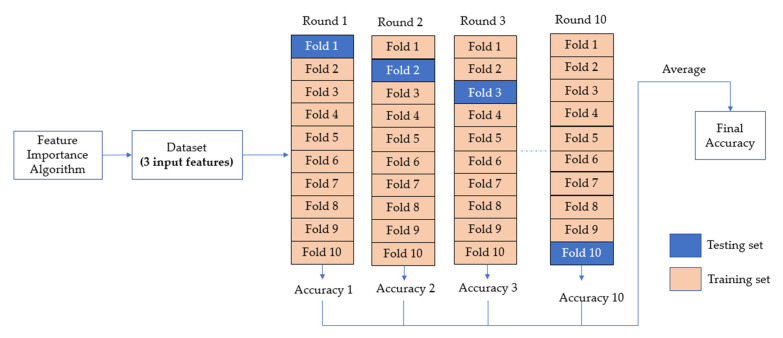
Feature optimization steps.

**Figure 6 ijerph-19-13962-f006:**
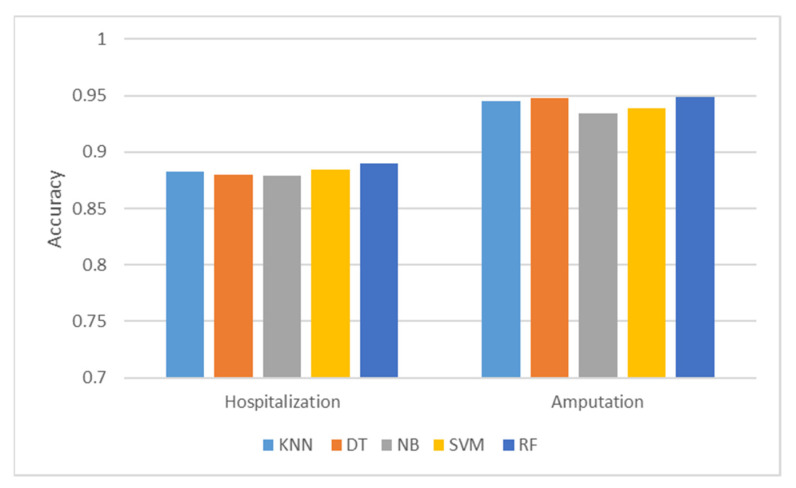
Accuracy Comparison of all ML models; K-Nearest Neighbors (KNN), Decision Tree (DT), Naïve Bayes (NB), Support Vector Machine (SVM), Random Forest (RF).

**Figure 7 ijerph-19-13962-f007:**
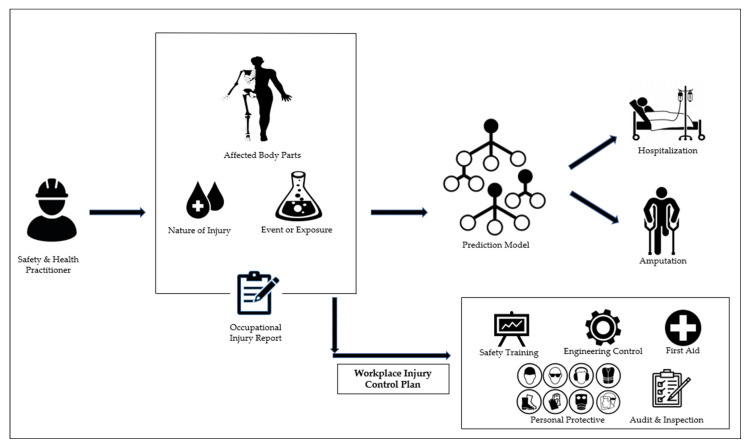
The proposed framework of AI-assisted occupational safety and health at workplace management.

**Table 1 ijerph-19-13962-t001:** Summary of ML Models of Occupational Injury Prediction in Existing Literature.

References.	Industry	Input Variables	ML Models	Findings
[18]	Construction	Age, sex, length of service, the type of construction, employer scale, and accident date.	LR, DT, RF, AdaBoost	RF is the best prediction model with the highest accuracy.
[19]	Construction	Year, type of work, type of accident, injured part, assailing materials, and cause of the accident.	SVM, Ensemble, PCA	SVM outperformed other models with higher accuracy in injury severity prediction.
[20]	Mining	Sub-unit, classification, accident type, occupation, activity, injury source, nature of the injury, injured body part.	DT, RF, ANN	ANN performed better than all other models.
[21]	Construction	15 variables: construction end use, event type, part of the body, cause of the accident (human and environment), and assigned tasks.	KNN, DT, RF	DT outperformed the other techniques with better sensitivity, recall, precision, and F1 score.
[22]	Construction	16 variables such as organization and behavior, technical management, resources support, management of the contract, safety training, and emergency management.	LR, DT, SVM, NB, KNN, RF, MLP, AutoML	NB and LR achieved good performance in F1-Score and AutoML is the best model to predict the severity of occupational injuries.

Note: Logistic Regression = LR, Decision Tree = DT, Random Forest = RF, Support Vector Machine = SVM, Naïve Bayes = NB, K-Nearest Neighbor = KNN, Artificial Neural Network = ANN, Principal Component Analysis = PCA, Multilayer Perceptron = MLP, AutoML = Automated Machine Learning.

**Table 2 ijerph-19-13962-t002:** Categorical variables and data distributions.

Variables	Categories	Distributions
Nature of Injury	N10 Traumatic injuries and disorders	2.3%
	N11 Traumatic injuries to bones, nerves, spinal cord	31.9%
	N12 Traumatic injuries to muscles, tendons, ligaments, joints	1.8%
	N13 Open wounds	34.2%
	N14 Surface wounds and bruises	1.1%
	N15 Burns and corrosions	5.4%
	N16 Intracranial injuries	3.6%
	N17 Effects of environmental conditions	2.6%
	N18 Multiple traumatic injuries and disorders	3.1%
	N19 Other traumatic injuries and disorders	14%
Type of Event	E01 Violence/other injuries by persons or animals	2.2%
	E02 Transportation incidents	8.5%
	E03 Fires and explosions	1.8%
	E04 Falls, slips, trips	30.4%
	E05 Exposure to harmful substances or environments	8.3%
	E06 Contact with objects and equipment	46.3%
	E07 Overexertion and bodily reaction	1.5%
	E09 Other(s)	1%
Source of Injury	S01 Chemicals and chemical products	2.8%
	S02 Containers, furniture, and fixtures	4.4%
	S03 Machinery	25.5%
	S04 Parts and materials	11.1%
	S05 Persons, plants, animals, and minerals	4.3%
	S06 Structures and surfaces	20.7%
	S07 Tools, instruments, and equipment	8.9%
	S08 Vehicles	14.1%
	S09 Other(s)	8.2%
Type of Industry	I11 Agriculture, Forestry, Fishing and Hunting	1.8%
	I21 Mining	2.9%
	I22 Utilities	1.3%
	I23 Construction	18%
	I31 Manufacturing	33%
	I42 Wholesale trade	5.6%
	I44 Retail trade	7.4%
	I48 Transportation and Warehousing	8.8%
	I51 Information	1%
	I52 Finance and Insurance	0.3%
	I53 Real Estate Rental and Leasing	1%
	I54 Professional, Scientific, and Technical Services	1.6%
	I55 Management of Companies and Enterprises	0.1%
	I56 Administrative/Waste Management and Remediation	5.6%
	I61 Educational Services	0.5%
	I62 Health Care and Social Assistance	4.7%
	I71 Arts, Entertainment and Recreation	1.3%
	I72 Accommodation and Food Services	2%
	I81 Other Services	1.9%
	I92 Public Administration	1.2%
Hospitalization	H1 Yes	80.6%
	H0 No	19.4%
Amputation	A1 Yes	26.4%
	A0 No	73.6%

**Table 3 ijerph-19-13962-t003:** Performance Prediction of all ML Models for Hospitalization.

ML Models	Accuracy	Precision	Recall	F1-score	AUC
KNN	0.883	0.957	**0.895**	0.925	0.86
DT	0.880	0.980	0.881	**0.928**	0.90
NB	0.879	**0.985**	0.862	0.920	**0.91**
SVM	0.884	0.984	0.870	0.924	**0.91**
RF	**0.890**	0.978	0.883	**0.928**	0.90

Note: Bold indicates the highest value on each performance metric.

**Table 4 ijerph-19-13962-t004:** Performance Prediction of all ML Models for Amputation.

ML Models	Accuracy	Precision	Recall	F1-Score	AUC
KNN	0.945	**0.869**	0.948	0.902	0.95
DT	0.948	0.861	0.959	0.907	0.95
NB	0.934	0.831	0.942	0.883	0.94
SVM	0.939	0.831	**0.967**	0.894	0.95
RF	**0.949**	0.860	0.963	**0.909**	0.95

Note: Bold indicates the highest value on each performance metric.

**Table 5 ijerph-19-13962-t005:** Feature Importance based on Ranking.

Feature	Importance Value	Description
Nature of Injury	0.406367	Identifies the main physical characteristic(s) of the occupational injury.
Type of Event	0.254030	Identifies how the occupational injury was produced.
Affected Body Part(s)	0.243115	Identifies the part of the body affected by the nature of the occupational injury.
Source of Injury	0.064195	Identifies the workplace factors such as objects, substances, equipment, and other external factors that were responsible for the occupational injury.
Type of Industry	0.032293	Identifies the nature of the organization, company, or enterprise

**Table 6 ijerph-19-13962-t006:** Cross-validation Scores.

Prediction	Criteria	Value
Hospitalization	Mean	0.893
	SD	0.0029
Amputation	Mean	0.949
	SD	0.0028

Note: SD = Standard Deviation.

**Table 7 ijerph-19-13962-t007:** Hyperparameter Tuning Performances.

Prediction	Criteria	Value
Hospitalization	Overall Accuracy	0.895
Amputation		0.954
Hospitalization Amputation	Optimized Parameters	‘n_estimators’: 1200,
		‘min_samples_split’: 15,
		‘min_samples_leaf’: 10,
		‘max_features’: ‘sqrt’,
		‘max_depth’: 15

## Data Availability

The link to publicly archived datasets for this study: https://www.osha.gov/severeinjury, last accessed on 25 July 2022.

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
