# Peer review of "Occupational Injury Risk Mitigation: Machine Learning Approach and Feature Optimization for Smart Workplace Surveillance"

_ijerph, 2022, doi:10.3390/ijerph192113962_

Round 1

Reviewer 1 Report

What do the values in Table 3 and Figure 1 represent? It is not clear to me what this mean and standard deviation mean, since they are categorical variables.

Figures 7 and 8 are not very relevant, since the ROC value was not decisive in choosing the best performance (RF). In fact Figure 8 is almost a repetition of the same graph.

Were data standardization or normalization techniques applied?

Page 10. Line 262: Why only the three most important features? What was the criteria?

In general I get that the work is interesting, but I consider that few input variables were included.

Author Response

Dear Reviewer 1, 

Thank you for your valuable comments given to improve the quality of the paper. We have extensively added all necessary work to ensure all comments have been addressed. 

Thank You. 

Reviewer 2 Report

This study predicts the hospitalization and amputation based on occupational injuries reports. The topic itself is of significance and authors organized the manuscript in a straightforward structure. However, there are several major design and presentation issues regarding its selection for descriptive statistics, input variables, output metrics. Manuscript's originality and innovation need to be more specific and highlighted.

1. Introduction, Page 2-3. As authors listing the relevant literatures in Table 1, they were all predicting the occupational injury outcomes using machine learning techniques and finding the best ML models. So what are the key differences between this study with those listed? Simply put it as "forecasting occupational injury outcomes is still lacking and restricted" or "generalizability of the findings and inadequate exploration of important variables" is somehow too vague to highlight the position of this study among literatures. Authors need to highlight their work's uniqueness.

2. Table 2 does not put Type of Industry as a variable while it is a categorical variables, right? You have shown its feature importance in Table 4.

3. Table 3 gives Mean, Median, and SD to describe their categorical variables such as Type of industry, nature of injury, source of injury, which does not align with the basic descriptive statistics knowledge and practical data understanding. Why use those statistics on categorical variables? In addition, why not give an appropriate statistics for the major target variables including hospitalization and amputation? If not correct or explain it, these parts really weakened the methodological soundness and readers trust on results.

4. Sections 2.3 and 2.4 introduces the predictive modeling methods and model evaluation metrics like a textbook way, it needs some linkage with this study such as customization of the models used in this study.

5. In Section 2.5, both testing and training datasets go through the k-fold validation process, right?

6. Sections 3.1 and 3.2 choose the best model based on Accuracy and F1-score. Would authors explain whether recall or precision is of high importance to look at in the context of this study objective?

7. Table 4. Feature value as a column name will be better to be replaced as importance value, right?

Author Response

Dear Reviewer 2, 

Thank you for your valuable comments given to improve the quality of the paper. We have extensively added all necessary work to ensure all comments have been addressed. 

Thank You. 

Reviewer 3 Report

This manuscript proposed a feature-optimized predictive model for anticipating occupational injury severity and studied the feature optimization technique that revealed the  important features. Overall, this paper is well organized.  Here are some suggestions for minor revisions.

Q1.  The key issue of this  manuscript is that the advanced machine learning methods are missed. Authors only employ the general machine learning methods, such as KNN, DT, NB, SVM, RF etc.  However, machine learning is hot topic and there are many new and useful models has been developed, Such as generative adversarial networks(GAN), deep learning etc.  It should be discussed or reviewed in Introduction. 

Q2. In Talbe 3. It seems RF model performs best in terms of 'Accuracy' and 'F1-Score'. It is better to discuss the reseaons in detail.

Q3. Fig.7 and Fig.8 are blurred. It is confusing. Please update using a limpid version.

Q4. Figure. 9 is  lack of standardization. What does vertical  axis stand for?  I guess it is 'Accuracy' and it should be marked.

Q5. 'AI+Medical Health' is a hot topic today and authors are suggested to give a more extensive review or discussion on this topic. There are many impressive works on this topic,  works using new ML methods in particular?  Such as:

Tensorizing GAN with high-order pooling for  Alzheimer's disease assessmentï¼›

Fine perceptive GANs for brain MR image super-resolution in wavelet domainï¼›

Diabetic Retinopathy Diagnosis using Multi-channel Generative Adversarial Network with Semi-supervision.

Q6: In table 5,  K-fold cross validation is used to evaluate the model performance.  It is better to give the mean and variance.

Author Response

Dear Reviewer 3, 

Thank you for your valuable comments given to improve the quality of the paper. We have extensively added all necessary work to ensure all comments have been addressed. 

Thank You. 
